# Korean Medicine Clinical Practice Guidelines for Lumbar Herniated Intervertebral Disc in Adults: Based on Grading of Recommendations Assessment, Development and Evaluation (GRADE)

**DOI:** 10.3390/healthcare10020246

**Published:** 2022-01-27

**Authors:** Bonhyuk Goo, Min-gi Jo, Eun-Jung Kim, Hyun-Jong Lee, Jae-Soo Kim, Dongwoo Nam, Jung Won Kang, Tae-Hun Kim, Yeon-Cheol Park, Yong-Hyeon Baek, Sang-Soo Nam, Myeong Soo Lee, Byung-Kwan Seo

**Affiliations:** 1Department of Acupuncture & Moxibustion, Kyung Hee University Hospital at Gangdong, Seoul 05278, Korea; goobh99@naver.com (B.G.); turtlessam@naver.com (M.-g.J.); 2Department of Acupuncture & Moxibustion Medicine, College of Oriental Medicine, Dongguk University, Gyeongju-si 38066, Korea; hanijjung@naver.com; 3Department of Acupuncture & Moxibustion Medicine, College of Korean Medicine, Daegu Haany University, Daegu 42158, Korea; whiteyyou@hanmail.net (H.-J.L.); jaice@daum.net (J.-S.K.); 4Department of Acupuncture & Moxibustion Medicine, College of Korean Medicine, Kyung Hee University, Seoul 02447, Korea; hanisanam@daum.net (D.N.); doctorkang@naver.com (J.W.K.); icarus08@hanmail.net (Y.-C.P.); byhacu@khu.ac.kr (Y.-H.B.); dangun66@gmail.com (S.-S.N.); 5Korean Medicine Clinical Trial Center, Department of Korean Medicine, Korean Medicine Hospital, Kyung Hee University, Seoul 02447, Korea; rockandmineral@gmail.com; 6KM Science Research Division, Korea Institute of Oriental Medicine, Daejeon 34054, Korea; drmslee@gmail.com

**Keywords:** clinical practice guideline, Korean medicine, lumbar intervertebral disc herniation, grading of recommendations assessment, development and evaluation

## Abstract

A significant number of individuals suffer from low back pain throughout their lifetime, and the medical costs related to low back pain and disc herniation are gradually increasing in Korea. Korean medicine interventions have been used for the treatment of lumbar intervertebral disc herniation. Therefore, we aimed to update the existing Korean medicine clinical practice guidelines for lumbar intervertebral disc herniation. A review of the existing guidelines for clinical treatment and analysis of questionnaires targeting Korean medicine doctors were performed. Subsequently, key questions on the treatment method of Korean medicine used for disc herniation in actual clinical trials were derived, and drafts of recommendations were formed after literature searches using the Grading of Recommendations, Assessment, Development and Evaluation. An expert consensus was reached on the draft through the Delphi method and final recommendations were made through review by the development project team and the monitoring committee. Fifteen recommendations for seven interventions for lumbar disc herniation were derived, along with the grade of recommendation and the level of evidence. The existing Korean medicine clinical practice guidelines for lumbar intervertebral disc herniation have been updated. Continuous updates will be needed through additional research in the future.

## 1. Introduction

With the ageing of the intervertebral disc, the compression force increases, squeezing out the nucleus pulposus through the fissure and resulting in the mechanical pressure on the spinal nerve that causes low back pain (LBP) and radiating pain, known as lumbar intervertebral disc herniation (LHIVD) [1].

In Korea, the overall annual incidence rate for spinal disease was a median of 16,387 per 100,000 individuals in 2016. The incidence rate and annual costs per patient increased by 7.6% and 14.7%, respectively, over the period from 2012 to 2016. The incidence and medical expenses of LHIVD were the highest in patients aged under 60 years [2].

Several clinical practice guidelines (CPGs) with a focus on Western medicine are available for LHIVD, such as CPGs for the diagnosis and treatment of LHIVD with radiculopathy (North American Spine Society, 2012) [3]. In Korea, several Korean medicine (KM) treatments are used to treat low back pain. As a result of a survey on low back pain patients conducted by the Ministry of Health and Welfare in 2017, 83.1% of outpatients and 90.3% of inpatients answered that they thought KM was effective, indicating a high level of trust in KM treatment [4].

In 2015, the Korea Institute of Oriental Medicine (KIOM) developed guidelines for LHIVD based on an acknowledgment of the need to amend the CPG evaluation instrument using the Appraisal of Guidelines for Research and Evaluation (AGREE) II [5]. We aimed to update the outdated 2015 KM CPGs for LHIVD and use Grading of Recommendations Assessment, Development and Evaluation (GRADE) to provide appropriate recommendations and address the key clinical questions about LHIVD in adults for KM doctors.

## 2. Materials and Methods

### 2.1. Planning of CPG Development

We collected and analyzed new data with previous data that was used as evidence in the latest KM CPGs for LHIVD [6], which was developed by KIOM in 2015. To establish a revision strategy, a questionnaire was developed to investigate KM doctors’ current utilization and opinions about KM CPGs for LHIVD. The survey was conducted by sending an email to all KM doctors. The survey items included awareness, utilization, accessibility, understanding, usability for explanation, usability for clinical decisions, and improvement points of the existing guidelines. To reflect the opinions of the clinicians, the frequency of intervention, demand for clinical trials, and use of herbal medicines were also investigated [7].

### 2.2. Development Process

A development committee analyzed previously developed CPGs for LHIVD and selected the key clinical questions with the results of the questionnaire survey. The Korean Acupuncture and Moxibustion Medicine Society, Society of Korean Medicine Rehabilitation, Association of the Spine and Joint Korean Medicine, and related experts reviewed and approved the key questions. After screening, quality evaluation, and synthesis of the retrieved literature, the recommendations were drafted. Expert consensus was reached on the draft using the Delphi method. After an internal review, an external review by the Korean Medical Standard Clinical Practice Guideline Project Group and the monitoring committee was conducted. Subsequently, the CPGs were reviewed and certified by the relevant group.

### 2.3. Establishment of the Expert Committee

The expert committee consisted of a working group and a review committee. The working group collected evidence on the key clinical questions and drafted the CPGs. The multidisciplinary experts, including economic evaluators, clinical experts, methodology experts, and guide users, participated in this activity. The review committee group reviewed the draft version and decided on the final recommendations.

### 2.4. Key Clinical Question

Acupuncture, pharmacopuncture, herbal medicine, Tuina manual therapy, moxibustion, thread-embedding acupuncture (TEA), and cupping were evaluated as KM interventions. The effects of single treatment, combination treatment, and differences in the effect according to the technique of the same treatment method were set for each intervention as three categories of clinical questions.

In the comparative intervention, the active control group included all interventions actually used for therapeutic purposes in Western medicine, and the category of conventional treatments applied as a combination therapy encompassed all the current KM and Western medicine interventions actually used for therapeutic purposes.

### 2.5. Search Strategy

The following international databases were used: PubMed, EMBASE, Cochrane Library, China National Knowledge Infrastructure, Citation Information by NII, and J-stage. The following Korean databases were used: KoreaMed, Korean Medical Database, Korean Studies Information Service System, National Digital Science Library, Korea Institute of Science and Technology Information, and Oriental Medicine Advanced Searching Integrated System. The search process was performed by setting the search period until May 2019, including the data from the last CPGs. Details regarding the search strategy and PICO approach are shown in the Appendix A.

### 2.6. Selection of Study

Two independent reviewers performed the screening procedure. Duplicate articles were excluded, and the selection and exclusion processes through the title, abstract, and full-text review were performed sequentially. The preferred reporting items for systematic reviews and meta-analyses (PRISMA) were adopted.

Randomized controlled clinical trial studies (RCTs) adopting the following three designs and including adult patients with LHIVD were selected based on the PICO approach:(1)KM treatment vs. active control.(2)KM treatment + conventional treatment vs. conventional treatment.(3)Specific technique of KM treatment vs. another technique of same KM intervention

The exclusion criteria were as follows:^1.^ Non-RCTs, such as literature reviews, case reports, observational studies, and animal experiments.^2.^ Unclear presentation of the evaluation tool or method.^3.^ In the case of an intervention that cannot be used in the clinical field or an intervention that cannot be viewed as a medical practice.^4.^ When the specific effect of the intervention alone cannot be confirmed due to the design problems of the experimental group and the control group.

### 2.7. Quality Assessment of the Studies

The Cochrane Risk of Bias Evaluation tool [8] was used for RCTs included in this study. All the evaluations were performed by two independent researchers. In case of disagreement, an agreement was reached with the aid of the supervisor.

### 2.8. Analysis and Synthesis of Evidence

The evidence gathered for each clinical question was synthesized and analyzed through a meta-analysis. The analysis was conducted using RevMan 5.3, provided by Cochrane.

The evidence was synthesized for each evaluation index of the evidence documents included in each clinical question. For a continuous variable, the mean difference (MD) was derived, and for a nominal variable, the risk ratio (RR) was derived to evaluate the magnitude and significance of the effect.

The magnitude and significance of the effect were used to establish a basic grade at the level of evidence and evaluate non-precision. The sample size of the data synthesized for each indicator was applied to the imprecision evaluation, and the heterogeneity data (I2) derived during the evidence synthesis process was applied to the inconsistency item when evaluating the level of evidence.

### 2.9. Classification of the Level of Evidence

We used the GRADE, developed by the Cochrane GRADE working group, to determine the level of evidence and the grade of recommendation [9]. In the GRADE, the level of evidence is preferentially determined according to the study design. RCTs are categorized as having a high level of evidence while observational studies are categorized as having a low level of evidence. If there is a risk of bias, inconsistency, indirectness, imprecision, or publication bias through the evaluation of evidence in a systematic literature review, the level of evidence is lowered by the first or second grade. If the effect size is large, the confounding variable reduces the effect size or if there is a dose–response relationship, the level of evidence can be increased.

The GRADE categorizes the level of evidence into four categories: high, moderate, low, and very low [10]. In the past, the definition of the level of evidence meant the possibility of change according to future research; however, since further research is not always possible, it has been revised to be the level of confidence now. In our CPGs, the classification of the classical text-based (CTB) level of evidence was also applied. The levels of evidence and definitions are provided in Table 1.

### 2.10. Development and Agreement of Recommendations

#### 2.10.1. Principles of Creating Recommendations

Recommendations were prepared according to the following principles:(1)The recommendations should contain a specific and accurate description of what management is appropriate for a particular situation and patient based on the evidence.(2)The key recommendations should be easily identifiable.(3)The recommendations and the supporting evidence should be linked.(4)The level of recommendation should be properly expressed.(5)The patients or population targeted for the recommendation and the recommended intervention should be specified in as much detail as possible.

#### 2.10.2. Recommendation Grade

The level of recommendation was determined by the magnitude of the benefit or harm. Based on the level of evidence, focusing on the level of confidence in the effect, comparison between the desired and unwanted effects, reliability of values and preferences, and use of resources was conducted.

When the benefit outweighs the harm, the use of the intervention is recommended and the recommendation grade A is assigned. According to the degree, grades B, C, and D are assigned separately.

If the CTB level of evidence is derived based on the textbook of KM prescribed by the Ministry of Health and Welfare and the Ministry of Food and Drug Safety and textbooks of the KM College, the utilization in the clinical field is evaluated and the Good Practice Point (GPP) grade was assigned through the expert consensus process of the development committee. An external consensus was conducted using the Delphi technique. The definitions and notations of the recommendation grades are summarized in Table 2.

### 2.11. AGREEII

According to the 23 main items of AGREE II, [11] the contents of CPGs was reviewed. Some points of errors, Such as Instances of essential content omission, non-specific content descriptions, descriptions of inappropriate contents were pointed out. A repeated feedback process was performed by adding and revising the inappropriate points.

## 3. Results

Our CPGs confirmed 15 recommendations based on the 7 types of KM treatments containing interventions, comparators, level of evidence, and grade of recommendation for the key clinical questions (Table 3). PRISMA flow charts and the GRADE data including the outcome measures, risk of bias, inconsistency, indirectness, imprecision, and other considerations are summarized in the Appendix A. The details of recommendation and clinical findings, including the number of patients, number of studies, and pooled relative effect with p values extracted from all the included studies, are summarized in the Appendix A.

### 3.1. Acupuncture

#### 3.1.1. Acupuncture vs. Active Control Treatment

The level of evidence and recommendations were derived based on 24 RCTs [12,13,14,15,16,17,18,19,20,21,22,23,24,25,26,27,28,29,30,31,32,33,34,35] comparing acupuncture for LHIVD and the active control treatment in terms of pain, function, and overall symptom improvement.

Acupuncture, electroacupuncture, and warm needle acupuncture were compared with active control treatments, such as Western medicine, injection, and physical therapy. The meta-analysis showed that acupuncture was more effective in improving overall symptoms (RR: 1.20, 95% confidence interval [CI]: 1.16–1.25, *p* < 0.001), pain (MD, 1.86, 95% CI: −1.91–−1.81, *p* < 0.001), and function (MD, 4.48, 95% CI: 3.93–5.03, *p* < 0.001) than active control treatment.

In conclusion, acupuncture is recommended for improving the overall symptoms of LHIVD(A/High).

#### 3.1.2. Acupuncture + Usual Conventional Therapy vs. Usual Conventional Therapy

The level of evidence and recommendations were derived based on 15 RCTs [20,36,37,38,39,40,41,42,43,44,45,46,47,48,49] that observed the combined effect of acupuncture and conventional treatment for LHIVD in terms of pain, function, and overall symptom improvement.

As a result of the meta-analysis, acupuncture combined with conventional treatments, such as moxibustion, herbal medicine, Western medicine, injection, and physical therapy, was more effective than usual conventional therapy in improving the overall symptoms (RR: 1.21, 95% CI: 1.16–1.28, *p* < 0.001), pain (MD, −1.03, 95% CI: −1.16–−0.90, *p* < 0.001), and function (ODI: MD, −3.27, 95% CI: −3.86–−2.68, *p* < 0.001; JOA: MD, 4.00, 95% CI: 3.48–4.52, *p* < 0.001). However, the heterogeneity (I2 = 80%) between studies was high; therefore, the level of evidence was evaluated at one level lower due to inconsistency. 

Acupuncture has been reported to have a low clinical risk in safety-related studies [50] and in the utilization survey of experts, it has been shown to be a highly useful treatment method [51].

In conclusion, acupuncture in combination with conventional treatment is recommended for improving the overall symptoms of LHIVD (A/Moderate).

#### 3.1.3. Electro-Acupuncture, Fire Needling, or Warm Needling vs. Acupuncture

The level of evidence and recommendations were derived based on seven RCTs [22,52,53,54,55,56,57] that observed the effect of additional thermal or electrical stimulation in acupuncture for LHIVD in terms of pain, function, and overall symptom improvement.

As a result of the meta-analysis, the addition of thermal stimulation or electrical stimulation during acupuncture was effective in improving the overall symptoms (RR: 1.16, 95% CI: 1.09–1.23, *p* < 0.001), pain (MD, −0.58, 95% CI: −0.76–−0.39, *p* < 0.001), and function (ODI: MD, −0.71, 95% CI: −1.29–−0.13, *p* < 0.05) compared to acupuncture monotherapy.

In conclusion, the addition of thermal stimulation or electrical stimulation during acupuncture is recommended for improving the overall symptoms of LHIVD (A/High).

#### 3.1.4. Deep-Insertion Acupuncture vs. Superficial-Insertion Acupuncture

The level of evidence and recommendations were derived based on eight RCTs [58,59,60,61,62,63,64,65] that observed the effect of the difference in the depth of insertion with respect to the pain and overall symptom improvement in acupuncture for LHIVD.

The meta-analysis showed that deep-insertion acupuncture was more effective in improving the overall symptoms (RR: 1.31, 95% CI: 1.23–1.39, *p* < 0.001) than superficial-insertion acupuncture. However, the level of evidence was evaluated to be one level lower due to inconsistency owing to the high heterogeneity (I2 = 88%) observed between studies. There was no significant difference in pain improvement (MD, −1.66, 95% CI: −3.97–0.65, *p* = 0.16).

In conclusion, deep-insertion acupuncture should be considered for improving the overall symptoms of LHIVD (B/Moderate).

### 3.2. Moxibustion

#### 3.2.1. Moxibustion vs. Active Control Group

As a result of a search for RCTs comparing moxibustion and active control treatment for LHIVD, one study [66] was found, but a sufficient sample size was not secured. There were no significant differences in terms of the effect (RR: 1.15, 95% CI: 0.97–1.36, *p* = 0.10).

In the classical literature contained in the textbooks of the College of KM [67], moxibustion is applied for cold back pain among the 10 classes of LBP. In the classical literature of Singugyeonglon, moxibustion is presented for waist and knee pain, which is similar to LBP and radiating pain. In addition, Donguibogam offers moxibustion for LBP, including when the waist cannot be bent or stretched with LBP.

In a questionnaire study that surveyed the treatment status of LHIVD among KM doctors, 102 of 373 respondents (27.3%) answered that they used moxibustion [7]. Additionally, in a survey on the current status of moxibustion for musculoskeletal disorders in KM doctors in Seoul, 135 of 234 respondents (57.7%) answered that they used moxibustion for LBP, indicating that the utilization of moxibustion in actual clinical practice is high [68].

The clinical evidence for the effect of moxibustion for LHIVD was found to be insufficient. However, based on the evidence from classical literature according to the development strategy of our guidelines, the level of evidence was assessed to be CTB. Considering the high utilization of moxibustio, the GPP grade was assigned through a clinical expert consensus process.

In conclusion, moxibustion is recommended for improving pain with LHIVD based on the consensus of the expert group (CTB/GPP).

#### 3.2.2. Moxibustion + Usual Conventional Therapy vs. Usual Conventional Therapy

The level of evidence and recommendations were derived based on five RCTs [47,57,69,70,71] that observed the combination effect of “moxibustion and acupuncture” or “moxibustion and Tuina manual therapy” for LHIVD in terms of pain, function, and overall symptom improvement.

As a result of the meta-analysis, the combination of moxibustion and conventional treatments, such as acupuncture and Tuina manual therapy, was more effective in improving the overall symptoms (RR: 1.22, 95% CI: 1.12–1.32, *p* < 0.001), pain (MD, −1.40, 95% CI: −1.85–−0.95, *p* < 0.001), and function (MD, 4.10, 95% CI: 3.42–4.77, *p* < 0.001) than usual conventional therapy, such as acupuncture or Tuina monotherapy.

In conclusion, moxibustion in combination with conventional treatment is recommended for improving the overall symptoms of LHIVD(A/High).

#### 3.2.3. Moxibustion Causing Deqi Sensation vs. Moxibustion Not Causing Deqi Sensation

The level of evidence and recommendations were derived based on four RCTs [72,73,74,75] that observed the effect of moxibustion for LHIVD in terms of function and overall symptom improvement.

The meta-analysis showed that moxibustion causing deqi sensation was effective in improving the overall symptoms (RR: 1.19, 95% CI: 1.06–1.33, *p* < 0.01) and function (MD, −2.66, 95% CI: −4.02–−1.30), *p* < 0.001) compared to moxibustion not causing deqi sensation. However, due to the lack of quality of the supporting literature and the number of subjects included, the level of evidence was evaluated at two levels lower due to the risk of bias and imprecision.

In conclusion, moxibustion treatment causing deqi sensation may be considered for improving the overall symptoms of LHIVD (C/Low).

### 3.3. Herbal Medicine

#### 3.3.1. Herbal Medicine vs. Active Control Treatment

The level of evidence and recommendations were derived based on seven RCTs [76,77,78,79,80,81,82] comparing herbal medicine and active control treatment for LHIVD in terms of pain, function, and overall symptom improvement.

The results of the meta-analysis showed that herbal medicine was more effective in improving the overall symptoms (RR: 1.19, 95% CI: 1.11–1.28, *p* < 0.001), pain (MD, −0.55, 95% CI: −0.70–−0.40, *p* < 0.001), and function (ODI: MD, −3.86, 95% CI: −4.71–−3.10, *p* < 0.001; JOA: MD, 1.46, 95% CI: 0.95–1.97, *p* < 0.001) than active control treatments, such as Western medicine and traction treatment.

Although the level of evidence was high in terms of overall symptom improvement, inconsistency and imprecision were observed in terms of pain and function improvement (VAS, I2 = 78%; ODI, I2 = 85%); therefore, the level of evidence was lowered by two grades.

In conclusion, herbal medicine should be considered to improve the overall symptoms of LHIVD (B/Moderate).

#### 3.3.2. Herbal Medicine + Usual Conventional Therapy vs. Usual Conventional Therapy

The level of evidence and recommendations were derived based on 22 RCTs [40,83,84,85,86,87,88,89,90,91,92,93,94,95,96,97,98,99,100,101,102,103] that observed the combination effect of herbal medicine and conventional treatment for LHIVD in terms of pain, function, and overall symptom improvement.

As a result of the meta-analysis, the combination of conventional herbal medicine with usual conventional treatments, such as acupuncture, pharmacopuncture, Tuina manual therapy, Western medicine, injection, physical therapy, and integrated treatment, was more effective in improving the overall symptoms (RR: 1.32, 95% CI: 1.26–1.37, *p* < 0.001), pain (MD, −1.51, 95% CI: −1.57–−1.46, *p* < 0.001), and function (ODI: MD, −5.25, 95% CI: −8.33–−2.17, *p* < 0.001; JOA: MD, 5.89, 95% CI: 5.31–6.46, *p* < 0.001) than usual conventional therapy.

In conclusion, herbal medicine in combination with conventional treatment is recommended for improving the overall symptoms of LHIVD (A/High).

### 3.4. Pharmacopuncture

#### Pharmacopuncture + Usual Conventional Therapy vs. Usual Conventional Therapy

The level of evidence and recommendations were derived based on eight RCTs [97,104,105,106,107,108,109,110] that observed the combined effect of pharmacopuncture and conventional treatment for LHIVD in terms of pain, function, and overall symptom improvement.

The meta-analysis showed that the combination of pharmacopuncture with conventional treatments, such as acupuncture and Tuina manual therapy, was more effective in improving the overall symptoms (RR: 1.19, 95% CI: 1.07–1.32, *p* < 0.001), pain (MD, –1.65, 95% CI: 1.70–−1.61, *p* < 0.001), and function (ODI: MD, −8.39, 95% CI: −10.50–−6.28, *p* < 0.001; ODI change: MD, 6.22, 95% CI: 3.10–9.33, *p* < 0.001; JOA: MD, 9.00, 95% CI: 7.89–10.11, *p* < 0.001) than usual conventional therapy. Since the overall number of evidence documents and the number of subjects included in studies was small, the level of evidence was lowered by one grade due to imprecision.

In conclusion, the combination of pharmacopuncture with conventional treatment should be considered for improving the overall symptoms of LHIVD (B/Moderate).

### 3.5. Tuina Manual Therapy

#### 3.5.1. Tuina Manual Therapy vs. Active Control Treatment

The level of evidence and recommendations were derived based on 10 RCTs [48,111,112,113,114,115,116,117,118,119,120] that compared Tuina manual therapy and active control treatment for LHIVD in terms of pain, function, and overall symptom improvement.

As a result of the meta-analysis, Tuina manual therapy was effective in improving the overall symptoms (RR: 1.17, 95% CI: 1.12–1.23, *p* < 0.001), pain (MD, 1.09, 95% CI: 1.32–−0.86, *p* < 0.001), and function (ODI: MD, 9.87; 95% CI: 15.68– –4.06; *p* < 0.001; JOA: MD, 4.85, 95% CI: 3.87–5.83, *p* < 0.001) compared to active control treatments, such as Western medicine, injection, and traction treatment.

In conclusion, Tuina manual therapy is recommended for improving the overall symptoms of LHIVD (A/High).

#### 3.5.2. Tuina Manual Therapy + Usual Conventional Therapy vs. Usual Conventional Therapy

The level of evidence and recommendations were derived based on 32 RCTs [36,39,46,89,99,121,122,123,124,125,126,127,128,129,130,131,132,133,134,135,136,137,138,139,140,141,142,143,144,145,146,147] that observed the combined effect of Tuina manual therapy and conventional treatment for LHIVD in terms of pain, function, and overall symptom improvement.

As a result of the meta-analysis, the combination of Tuina manual therapy with conventional treatments, such as acupuncture, herbal medicine, injection, and traction therapy, was more effective in improving the overall symptoms (RR: 1.25, 95% CI: 1.22–1.29, *p* < 0.001), pain (MD, −1.08, 95% CI: 1.21–−0.95, *p* < 0.001), and function (ODI: MD, 2.93, 95% CI: 3.38 –−2.49, *p* < 0.001; JOA: MD, 4.86, 95% CI: 4.19–5.53, *p* < 0.001) than conventional treatment. However, the heterogeneity (I2 = 76%) between studies was high; therefore, the level of evidence was evaluated at one level lower due to inconsistency.

In conclusion, a combination of Tuina manual therapy and conventional treatment is recommended for improving the overall symptoms of LHIVD (A/Moderate).

### 3.6. TEA

#### 3.6.1. TEA vs. Active Control Treatment

The level of evidence and recommendations were derived based on 12 RCTs [148,149,150,151,152,153,154,155,156,157,158,159] that compared TEA and active control treatment for LHIVD in terms of pain, function, and overall symptom improvement.

As a result of the meta-analysis, TEA was more effective in improving the overall symptoms (RR: 1.14, 95% CI: 1.10–1.19, *p* < 0.001), pain (MD, 0.40, 95% CI: 0.54–−0.26, *p* < 0.001), and function (ODI: MD, 1.30, 95% CI: 2.42–−0.18, *p* < 0.05; JOA: MD, 2.03, 95% CI: 0.30–3.76, *p* < 0.05) than active control treatments, including acupuncture and complex treatment. However, in terms of overall symptom improvement, the risk of bias was high; therefore, the level of evidence was lowered by one grade.

In conclusion, TEA should be considered to improve the overall symptoms of LHIVD (B/Moderate).

#### 3.6.2. TEA + Usual Conventional Therapy vs. Usual Conventional Therapy

The level of evidence and recommendations were derived based on seven RCTs [160,161,162,163,164,165,166] that observed the combined effect of TEA and conventional treatment in terms of pain, function, and overall symptom improvement for LHIVD

As a result of the meta-analysis, the combination of TEA and conventional treatments, such as acupuncture, herbal medicine, and traction treatment, was more effective in improving the overall symptoms (RR: 1.15, 95% confidence interval [CI] 1.09–1.21, *p* < 0.001), pain (MD, −2.00, 95% CI: –2.46–−1.54, *p* < 0.001), and function (ODI: MD, 21.07, 95% CI: 27.18–−14.96, *p* < 0.001; JOA: MD, 2.37, 95% CI: 0.78–3.96, *p* < 0.001) compared to usual conventional therapy. However, in the area of overall symptom improvement, heterogeneity (I2 = 80%) between studies was observed, and the level of evidence was lowered by one level due to inconsistency.

In conclusion, TEA in combination with conventional treatment should be considered for improving the overall symptoms of LHIVD (B/Moderate).

### 3.7. Cupping

#### Cupping + Usual Conventional Therapy vs. Usual Conventional Therapy

The level of evidence and recommendations were derived based on five RCTs [167,168,169,170,171] that observed the combined effect of cupping and conventional treatment for LHIVD in terms of pain, function, and overall symptom improvement.

As a result of the meta-analysis, the combination of cupping and conventional treatment had a significant effect on the overall symptom improvement (RR: 1.43, 95% CI: 1.27–1.62, *p* < 0.001) compared to conventional therapy without cupping. However, there was no significant difference in pain improvement (MD, −1.08, 95% CI: −2.24–0.08, *p* = 0.07); therefore, the level of evidence was evaluated as very low due to the high imprecision.

In conclusion, cupping treatment should be considered in combination with conventional treatment for improving the overall symptoms of LHIVD (B/Moderate).

## 4. Discussion

CPGs are systematically developed statements to assist practitioners and patient decisions about the appropriate healthcare for specific clinical circumstances.

Among the several standard methods used to develop CPGs, we mainly used the GRADE to assess the quality of evidence. 

We applied seven types of interventions to the clinical question. In a preliminary study, it was found that the studied interventions were used frequently in the actual clinical field [7]. The studied interventions are often used alone or in combination with other treatments.

When the intervention was applied alone, acupuncture and Tuina manual therapy were evaluated as A grade, herbal medicine and TEA were evaluated as B grade, and moxibustion was evaluated as GPP grade. Each single treatment was compared to active control treatments, including drugs, injection therapy, and physical therapy. This comparison showed that KM treatment can be used as an alternative to conventional treatment.

For herbal medicine, the level of evidence was lowered by two grades due to the inconsistency and imprecision observed in terms of pain and function improvement. For TEA, the level of evidence was lowered by one grade due to the risk of bias. Further research is required to expand the evidence.

When intervention was applied as combination therapy, the combination of acupuncture with active control treatment, combination of moxibustion with acupuncture or Tuina manual therapy, combination of herbal medicine with active control treatment, and combination of Tuina manual therapy with active control treatment were evaluated as A grade, and the combination of pharmacopuncture with active control treatment, combination of TEA with active control treatment, and combination of cupping treatment with active control treatment were evaluated as B grade. Conventional therapies include Western medicine and KM treatments. This comparison showed that KM treatment can be used as a complementary treatment to conventional treatments.

Regarding the combination of pharmacopuncture with active control treatment, the level of evidence was lowered by one grade due to imprecision because the overall number of evidence documents and the number of subjects included in the studies were small. For the combination of Tuina manual therapy with active control treatment and combination of TEA with active control treatment, the level of evidence was evaluated as one level lower due to the inconsistency owing to the high heterogeneity between studies. Further research is required to expand the evidence.

In actual clinical practice, KM techniques have different treatment techniques. Among them, we developed recommendations for the depth of acupuncture, thermal and electrical stimulation with acupuncture, and deqi sensation caused by moxibustion. Additional heat or electrical stimulation during acupuncture was classified as grade A. Deep-insertion acupuncture was evaluated as B grade, and moxibustion causing deqi sensation was evaluated as C grade. For deep-insertion acupuncture, the level of evidence was evaluated as one level lower due to inconsistency owing to the high heterogeneity between studies. Regarding moxibustion that causes deqi sensation, the level of evidence was evaluated at two levels lower due to the risk of bias and imprecision owing to the lack of quality of the supporting data and the n umber of subjects included. Further research is required to expand the evidence.

### 4.1. Limitations of the Present Guidelines

Our CPGs have several limitations. First, there are limitations in the search strategy. Since the terms were not the same for each treatment intervention, it was difficult to present a standardized method for selecting a search word, and there were limitations in establishing a consistent level of search strategy due to the different terms by country.

Second, there are some qualitative limitations that include cases where bias risk evaluation factors were not presented, cases that were not blinded, and cases where there were limitations in the design of the study. Moreover, some studies had limitations, such as study inconsistency and imprecision, and a lack of evaluation indicators, such as segmentation, safety, and economics.

Finally, there are methodological limitations that do not reflect the diversity of KM treatment techniques. Few studies have compared and analyzed the detailed elements of KM treatments. There may be some disparity from the actual clinical practice in the method of synthesizing KM treatments, including various detailed attributes, into a certain category and arriving at a conclusion. Except for the diagnosis of LHIVD, there is insufficient evidence to consider clinical variables, such as severity.

### 4.2. Recommendation for Further Guidelines

To overcome the limitations of our CPGs, a number of clinical studies are needed to accumulate evidence. In addition, to acquire data on stability and provide recommendations that can be realistically applied in the clinical field, it is necessary to expand the scope to case reports and observational studies while using RCTs as supporting literature in CPGs. A close review of more clinical experts is necessary to prevent the deterioration of quality.

## Figures and Tables

**Table 1 healthcare-10-00246-t001:** GRADE level of evidence.

Level of Evidence	Definition
High	We are very confident that the true effect lies close to that of the estimate of the effect.
Moderate	We are moderately confident of the effect estimate; the true effect is likely to be close to the estimate of the effect; however, there is a possibility that it is substantially different.
Low	Our confidence in the effect estimate is limited: The true effect may be substantially different from the estimate of the effect.
Very low	We have very little confidence in the effect estimate; the true effect is likely to be substantially different from the estimate of the effect.
Classical text-based	Although there is evidence recorded in the classical texts, such as traditional Korean medicine books, evidence studies using modern research methodology have not been conducted.

**Table 2 healthcare-10-00246-t002:** GRADE definitions and notations.

Grade	Definition	Notation
A	It is recommended when the benefits are clear, and utilization is high in the clinical field.	Is recommended
B	It is given when the benefit is reliable, and the utilization is high or moderate in the real-world practice or when the clinical benefit is obvious even though the evidence from the research related to the evidence of the recommendation is insufficient.	Should be considered
C	It is given when the benefit is not reliable; however, the utilization is high or moderate in the treatment field.	May be considered
D	Benefits are unreliable and can cause harmful consequences.	Is not recommended
Good Practice Point	It is recommended on the basis of a group of experts based on bibliographic evidence or clinical utilization.	Is recommended based on the expert group consensus

**Table 3 healthcare-10-00246-t003:** Interventions, comparators, and level of evidence/grade of recommendation for the key clinical questions.

Key Clinical Question	Intervention (I)	Comparator (C)	Grade of Recommendation/Level of Evidence/
**Acupuncture**			
CQ1. Is acupuncture treatment helpful to improve the overall symptoms in adult patients with lumbar intervertebral disc herniation compared to usual conventional treatment (UCT)?	Acupuncture	UCT	A/High
CQ2. Is acupuncture treatment with UCT helpful to improve the pain and overall symptoms in adult patients with lumbar intervertebral disc herniation compared to UCT?	Acupuncture + UCT	UCT	A/Moderate
CQ3. Is electrio-acupuncture, fire needling, or warm needling treatment helpful to improve the overall symptoms in adult patients with lumbar intervertebral disc herniation compared to acupuncture?	Electro-acupuncture, fire needling, orwarm needling	Acupuncture	A/High
CQ4. Is the deep-injection acupuncture treatment helpful to improve the overall symptoms in adult patients with lumbar intervertebral disc herniation compared to superficial-injection acupuncture?	Deep-injection acupuncture	Superficial-injection acupuncture	B/Moderate
**Moxibustion**			
CQ5. Is moxibustion treatment helpful to improve the overall symptoms in adult patients with lumbar intervertebral disc herniation compared to UCT?	Moxibustion	UCT	GPP/Very low
CQ6. Is a combination treatment or moxibustion with acupuncture or Tuina manual therapy treatment helpful to improve the overall symptoms in adult patients with lumbar intervertebral disc herniation compared to acupuncture or Tuina monotherapy?	Moxibustion + acupuncture or moxibustion + Tuina manual therapy	Acupuncture or Tuina manual therapy	A/High
CQ7. Is moxibustion treatment that suspends deqi sensation helpful in improving the overall symptoms in adult patients with lumbar intervertebral disc herniation compared to moxibustion treatment that does not cause deqi sensation?	Moxibustion with deqi sensation	Moxibustion without deqi sensation	C/Low
**Herbal medicine**			
CQ8. Is herbal medicine monotherapy helpful to improve the overall symptoms in adult patients with lumbar intervertebral disc herniation compared to UCT?	Herbal medicine	UCT	B/Moderate
CQ9. Is herbal medicine with UCT helpful to improve the overall symptom in adult patients with lumbar intervertebral disc herniation compared to UCT?	Herbal medicine + UCT	UCT	A/High
**Pharmacopuncture**			
CQ10. Is pharmacopuncture treatment with UCT helpful to improve the overall symptoms in adult patients with lumbar intervertebral disc herniation compared to UCT?	Pharmacopuncture + UCT	UCT	B/Moderate
**Tuina manual therapy**			
CQ11. Is Tuina manual therapy helpful to improve the overall symptoms in adult patients with lumbar intervertebral disc herniation compared to UCT?	Tuina manual therapy	UCT	A/High
CQ12. Is Tuina manual therapy with UCT helpful to improve the overall symptoms in adult patients with lumbar intervertebral disc herniation compared to UCT?	Tuina manual therapy + UCT	UCT	A/Moderate
**Thread-embedding acupuncture**			
CQ13. Is thread-embedding acupuncture monotherapy helpful to improve the overall symptoms in adult patients with lumbar intervertebral disc herniation compared to UCT?	Thread-embedding acupuncture	UCT	B/Moderate
CQ14. Is thread-embedding acupuncture therapy with UCT helpful to improve the overall symptoms in adult patients with lumbar intervertebral disc herniation compared to UCT?	Thread-embedding acupuncture + UCT	UCT	B/Moderate
**Cupping**			
CQ15. Is cupping with UCT helpful to improve the overall symptoms in adult patients with lumbar intervertebral disc herniation compared to UCT?	Cupping + UCT	UCT	B/Moderate

UCT: usual conventional treatment.

## Data Availability

All relevant data are included in this manuscript and Appendix A.

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
