# Peer review of "Korean Medicine Clinical Practice Guidelines for Lumbar Herniated Intervertebral Disc in Adults: Based on Grading of Recommendations Assessment, Development and Evaluation (GRADE)"

_healthcare, 2022, doi:10.3390/healthcare10020246_

Round 1

Reviewer 1 Report

Very thorough study and very well done. Some grammatical corrections are needed but the content is very good and I feel this will make a good contribution to the Korean Medical Literature and practice guidelines. 

I reviewed the supplementary files and I see how much work went into this project. If some of the supplementary material could make its way into the manuscript, like forrest plots comparing the interventions. 

The are grammar errors throughout, and I believe the periods should follow the in-text citations in brackets.

Table-1 needs reformatting, perhaps the figure/table didn't upload properly when the manuscript was submitted

I like how the authors included comparison of superficial vs deep acupuncture for LHIVD.

Author Response

First of all, thank you for your kind and accurate review opinion.
As you said, forest plots have been added. Due to the size problem, it will be added to the supplementary material.
Also, the period is rewritten after brackets, and some grammatical errors have been corrected. Finally, the table-1 form has been modified. Once again, thank you for your comments, and please see the attached file for details of changes.  

Reviewer 2 Report

This paper presents a new and updated clinical practice guideline of Korean medicine for lumbar intervertebral disc herniation. The Grading of Recommendations, Assessment, Development and Evaluation (GRADE) was used to draft the recommendations. The consensus among the experts was conducted using the Delphi method, and the final recommendations were developed through review by the development project team and the monitoring committee. In total, fifteen recommendations were proposed for seven lumbar disc herniation interventions with their respective recommendation grade and evidence level. The study presents progress in the area, but minor points are to be corrected.

INTRODUCTION

  1. If the report contains data for 2012 to 2016, why is the incidence rate for spine reported for 2014 and not 2016? Line 44.
  2. It is suggested to highlight the reasons for creating this new clinical practice guideline of Korean medicine for lumbar intervertebral disc herniation. Is the guideline developed by the Korea Institute of Oriental Medicine (KIOM) flawed or outdated?

METHOD

  1. The authors adequately describe and provide sufficient information on the development process, critical clinical question, selection of study, quality assessment of the studies, classification of the level of evidence and recommendation. However, ¿why not report the AGREE II procedure and results? It is suggested to describe and report the AGREE II performed.

RESULTS

  1. The authors clearly presented the results.
  2. The results have been described in inverted form in Table 3 (i.e, the column with the levels of evidence and the grade of recommendation). It is suggested to modify the order of presentation of the results in the column levels of evidence/degree of recommendation.
  3. In line 231, the subtitle “Acupuncture + usual conventional therapy vs acupuncture” is centred and not on the left side, and it is suggested to align to the left.

DISCUSSION

  1. The authors adequately discuss the results obtained and justify their limitations. Also, they stated the implications of their findings. 

Author Response

First of all, thank you for your kind and accurate review opinion.

As you said, the incidence rate for spine reported has been revised as of 2016.

Also, latest KM LHIVD CPG developed by KIOM in 2015 need to update due to

it has outdated data.

AGREE II procedure and results newly added to files. Due to the size problem, AGREE II checklist will be added to the supplementary material.

Also, the period is rewritten after brackets, and some grammatical errors have been corrected.

Finally, the order of Table 3 has been modified and  the subtitle “Acupuncture + usual conventional therapy vs acupuncture” is aligned to the left.

Once again, thank you for your comments, and please see the attached file for details of changes.